# Analyzing the Features Affecting the Performance of Teachers during Covid-19: A Multilevel Feature Selection

Alqahtani Saeed [1], Raja Habib [2], Maryam Zaffar [2,*], Khurrum Shehzad Quraishi [3], Oriba Altaf [2], Muhammad Irfan [4], Adam Glowacz [5], Ryszard Tadeusiewicz [6], Mohammed Ayed Huneif [7], Alqahtani Abdulwahab [7], Sharifa Khalid Alduraibi [8], Fahad Alshehri [8], Alaa Khalid Alduraibi [8] and Ziyad Almushayti [8]

[1] Department of Surgery, Faculty of Medicine, Najran University, Najran 61441, Saudi Arabia; alhafezsaeed@gmail.com
[2] Department of Computer Science and Information Technology, University of Lahore, Islamabad 44000, Pakistan; raja.habib@se.uol.edu.pk (R.H.); Aribakahan@gmail.com (O.A.)
[3] Department of Process Engineering, Pakistan Institute of Engineering & Applied Sciences, Nilore, Islamabad 44000, Pakistan; qureshiksq@pieas.edu.pk
[4] Electrical Engineering Department, College of Engineering, Najran University Saudi Arabia, Najran 61441, Saudi Arabia; miditta@nu.edu.sa
[5] Department of Automatic Control and Robotics, Faculty of Electrical Engineering, Automatics, Computer Science and Biomedical Engineering, AGH University of Science and Technology, aleja Adama Mickiewicza 30, 30-059 Kraków, Poland; adglow@agh.edu.pl
[6] Department of Biocybernetics and Biomedical Engineering, Faculty of Electrical Engineering, Automatics, Computer Science and Biomedical Engineering, AGH University of Science and Technology, aleja Adama Mickiewicza 30, 30-059 Kraków, Poland; rtad@agh.edu.pl
[7] Department of Pediatrics, College of Medicine, Najran University, Najran 61441, Saudi Arabia; huneif@hotmail.com (M.A.H.); aaalsharih@nu.edu.sa (A.A.)
[8] Department of Radiology, College of Medicine, Qassim University, Buraidah 52571, Saudi Arabia; dr.s.alduraibi@gmail.com (S.K.A.); f.alshehri@qu.edu.sa (F.A.); AI.alderaibi@qu.edu.sa (A.K.A.); ziyadalmushayti@qu.edu.sa (Z.A.)
* Correspondence: maryam.zaffar82@gmail.com

**Abstract:** COVID-19 is a profoundly contagious pandemic that has taken the world by storm and has afflicted different fields of life with negative effects. It has had a substantial impact on education which is evident from the transition from Face-to-Face (F2F) teaching to online mode of education and the rigid implementation of lockdown across the globe. This study examines the factors impacting the performance of teachers during the lockdown period of COVID-19 using various feature selection algorithms and Artificial Intelligence techniques. In this paper, we have proposed a novel multilevel feature selection for the prediction of the factors affecting the teachers' satisfaction with online teaching and learning in COVID-19. The proposed multilevel feature selection is composed of the Fast Correlation Based Filter (FCBF), Mutual Information (MI), Relieff, and Particle Swarm Optimization (PSO) feature selection. The performance of the proposed feature selection approach is evaluated through the teachers' benchmark dataset. We used a range of measures like accuracy, precision, f-measure, and recall to evaluate the performance of the proposed approach. We applied different machine learning approaches (SVM, LGBM, and ANN) with the proposed multilevel feature selection technique. The performance of the proposed approach is also compared with existing feature selection algorithms, and the results show the improvement in the performance of feature selection in terms of accuracy, precision, recall, and F-Measure. Proposed feature selection provides more than 80% accuracy with Light Weight Gradient Boosting Machine (LGBM).

**Keywords:** feature selection; teachers; COVID-19; educational data mining; machine learning

## 1. Introduction

The COVID-19 pandemic has completely transformed life as we knew it. The outbreak of this pandemic caught the world unprepared and by surprise. The virus continues to wreak havoc resulting in new infections and deaths. This virus had a severe impact on various facets of life such as the economy, jobs, tourism, and sports, etc. The education department was not exempt from this pandemic either. Some governments enforced stern lockdown in the aftermath of this predicament. As a part of this lockdown, the educational institutions including schools, colleges, and universities were shut down and the transition from Face-to-Face education to online learning came about.

Online education has emerged as an unavoidable alternative for educational institutions during the COVID-19 outbreak and it has helped the students to continue with their education despite the COVID-19 pandemic [1,2]. To make online education a success, teachers and students adapted to cope with new technology.

The positive attitude of teachers towards new technologies (zoom, MS Teams, etc.) helped in continuing the educational process smoothly. Teachers have been putting in incessant effort to deliver online lectures efficaciously. However, the satisfaction of teachers plays a vital role in delivering adept online lectures too. Existing studies define instructor satisfaction as the assumption that the process of online teaching is efficient, effective, and beneficial for the stakeholders. The importance of faculty satisfaction can be validated from the fact that the Sloan Consortium factors it in as one of the five pillars in the quality framework for online education [3]. The satisfaction of teachers directly impacts the satisfaction of students, as a satisfied teacher delivers an online lecture, and motivates students more positively.

Recent studies during COVID-19 analyzed the factors affecting the satisfaction of teachers teaching at different levels (primary, secondary, lower secondary, and post-secondary classes). Gender, teachers' experience, technical support to teachers are factors identified by the recent studies playing a vital role in the satisfaction of teachers during COVID-19. To the best of our knowledge, no existing study applied feature selection techniques on the pre-processing step of COVID-19 teachers' datasets.

In this research, we have attempted to investigate and evaluate the factors that impact the quality of education during the COVID-19 pandemic. The study reports the factors that have an impact on the satisfaction of faculty as well as their perceptions towards opportunities and challenges of online learning. This could very well be the first study that makes use of multiple-level feature selection and machine learning algorithms for this purpose. The results of the study could be useful in elevating the quality of online education and making it a success during the COVID-19 pandemic situation. The findings of the study could be important in designing and planning various faculty development programs. We have proposed multiple-level feature selection for selecting the factors affecting the teacher's satisfaction with online teaching during the COVID-19 pandemic. Our research utilized the Vietnam teacher's dataset of 294 different level teachers with 36 features. The COVID-19 triggers academic setbacks and a lot of negative impacts on educational stakeholders. No doubt E-learning has an advantage with the prospect of learning from any place in the world, and it helps in the continuation of the learning process during COVID-19 pandemic lockdown situations. This quick digital transformation of learning from Face-to-Face to E-learning affects the performance of teachers and students. Countries all over the world faced a lot of difficulties in digital transformation. In our research, we have selected the Vietnam teacher dataset for analysis, as Vietnam was also not well prepared for digital transformation, due to limited technology and no awareness of a controlled e-learning system in terms of quality teaching and assessment [1].

The main aim of the study is to evaluate the COVID-19 related teachers' dataset to identify the factors affecting the satisfaction of teachers on online learning during COVID-19. The proposed approach is explained in detail in Section 3. The main contributions of the proposed work are as follows:

- Identifying the factors affecting the satisfaction of teachers with online teaching and learning during COVID-19.
- Provide awareness about the factors affecting the teacher's performance during COVID-19 to take proactive decisions for the improvement of the quality of education during COVID-19.
- Proposing multiple-level feature selection for selecting the factors affecting the teacher's satisfaction with online teaching during the COVID-19 pandemic.
- Explore traditional machine learning approaches namely SVM, LGBT, and ANN for selecting the optimal feature set affecting the satisfaction of teachers with online teaching during COVID-19.

The remainder of this paper is structured as follows: Section 2 provides a brief overview of the related literature. Section 3 provides the details of the proposed multilevel feature selection approach; Section 4 presents the results of the proposed approach, and Section 5 concludes the research and future work.

## 2. Related Work

During the COVID-19 emergency, numerous nations shut down the schools to restrict the spread of the infection [2], and this emergency leads to the issue of on-time adoption of new technology for the teaching and learning process [3]. The online teaching and learning process put the learners and teachers in a very challenging situation [4]. In addition, instructors have confronted significant difficulties carrying out internet learning, for example, advanced disparity among student marks, poor and inadmissible substance and instructional materials, the shortfall of help and preparing, and the issue of showing quality [5,6]. Despite unforeseen conditions during online teaching in COVID-19 teachers try their best to help the students in their studies, educators have discovered advanced devices to convey instructions to their students and coordinate correspondence inside their classes [7,8]. Teachers adopted a new methodology of teaching to cope with the teaching and learning issues during pandemic situations [9].

During COVID-19, different universities even changed the evaluation criteria like projects, remarkable tasks, and constant evaluations as final evaluation [10–15]. Teachers teaching at a different level of education experience different issues while online teaching and learning. Teachers are the main stakeholders of the educational process; it is very important to figure out the factors affecting their performance. Especially in the COVID-19 pandemic, as lockdown situations change the norms of the educational process, so the possibility of different challenges comes in front of educators. Different studies are conducted to evaluate the challenges faced by the teachers in COVID-19 and an overview of recent studies is presented in Table 1. Satisfaction with online learning is a substantial aspect of boosting successful educational activities and improving the quality of education [16].

**Table 1.** Overview of Recent studies on the effect of COVID-19 on Teachers.

| Paper | No Teachers/ Size of Dataset | Level of Teachers | Objective | Results/Factors |
|---|---|---|---|---|
| [16] 2021 | 81 | Medical and health science college teachers | Identifying the factors affecting the satisfaction of teachers | Technical assistance during online learning in COVID-19 may enhance the satisfaction of teachers. |
| [17] 2020 | 45 | Elementary school teachers | Effect of online learning on elementary teachers in COVID-19. | 80% of teachers feel dissatisfaction with online learning in COVID-19 |
| [18] 2021 | 98 | Public School Mathematics teachers | Teacher's attitude towards online teaching in COVID-19 | Male Mathematics teachers showed more positivity in terms of new technology for teaching mathematics than female teachers in the COVID-19 pandemic. |

**Table 1.** *Cont.*

| Paper | No Teachers/ Size of Dataset | Level of Teachers | Objective | Results/Factors |
|---|---|---|---|---|
| [19] 2020 | 27 | English as a Foreign Language (EFL) teachers | Challenges faced by teachers during COVID-19 | Online teacher education affects the performance of teachers during online learning in COVID-19 |
| [20] 2020 | 55 | Junior and senior high school teachers | Problems faced by teachers during learning in COVID-19 | Technical support from school helps the teachers in conducting online classes in COVID-19. |
| [21] 2020 | 18 | Vocational school teachers | Identify challenges for teachers in COVID-19 | The continuation of the teaching and learning process is due to the positive response of teachers towards adopting new technologies for online learning in COVID-19. |
| [22] 2020 | 643 | School teachers | Factors affecting the adoption of online teaching during COVID-19 | Length of teaching service plays a vital role in shaping the attitude of teachers towards adopting new skills of online teaching during COVID-19 |
| [23] 2021 | 239 | University teachers | Effect of adopting new norms of teaching methods on teacher's performance during COVID-19 | Young teachers adopt the new technology more positively, whereas organizational ignorance also affects the performance of teachers |
| [24] | 670 | University Teachers | Effect of COVID-19 on teachers | Online teaching during COVID-19 affects negatively on teacher's performance |
| [25] | 307 | Expat teachers of different levels | Factors affecting ex-pat teacher during COVID-19 | Expat teachers mostly intended to leave during COVID-19 due to anxiety |

Table 1 gives a brief view of the studies conducted on the identification of factors affecting teachers during COVID-19. Table 1 shows that recent studies are conducted on the various number of teachers teaching at a different level of education. Different factors are identified in recent studies that may affect the performance of the teachers during online teaching in COVID-19. Gender, teaching experience, support of the administration, awareness of new technology are the factors mostly identified by the recent studies. Furthermore, as the recent studies pointed the issue of unsatisfaction of teachers with online learning during COVID-19, motivates us in identifying the factors affecting the satisfaction of teachers with online teaching and learning. Moreover, there is a need to utilize AI techniques for the identification of factors affecting the performance of teachers in COVID-19. The proposed feature selection techniques are presented in detail in Section 3.

## 3. Proposed Methodology

Figure 1 illustrates the proposed methodology to figure out the factors affecting the performance of the teachers during online learning in COVID-19. The proposed approach is evaluated through 294 Vietnamese teachers' datasets during COVID-19 [7], collected through different social media sources. Different filter feature selection algorithms are hybridized to extract optimal features from the COVID-19 teacher dataset. There are mainly two types of feature selection algorithms, filter feature selection, and wrapper feature selection. Filter feature selection considers the feature relation with target feature/class as well as feature relation with other features in a dataset. Whereas wrapper feature selection mainly considers the relation of feature with target feature/class. The advantage of filter in terms of taking into consideration the importance of each of the features in the dataset leads us to utilize filter feature selection algorithms in our proposed methodology presented in this paper. As teachers are the main stakeholders of the teaching and learning process, so

it is important to figure out what are the factors that may affect the teacher's satisfaction towards online teaching and learning during COVID-19.

This paper will help the educational administrators as well as countries' government officials to take proactive measures to improve the performance of teachers, during online teaching and learning. The improved performance of teachers may also affect students' performance. The reason for multilevel feature selection is to obtain the most relevant factors affecting the performance of teachers during COVID-19.

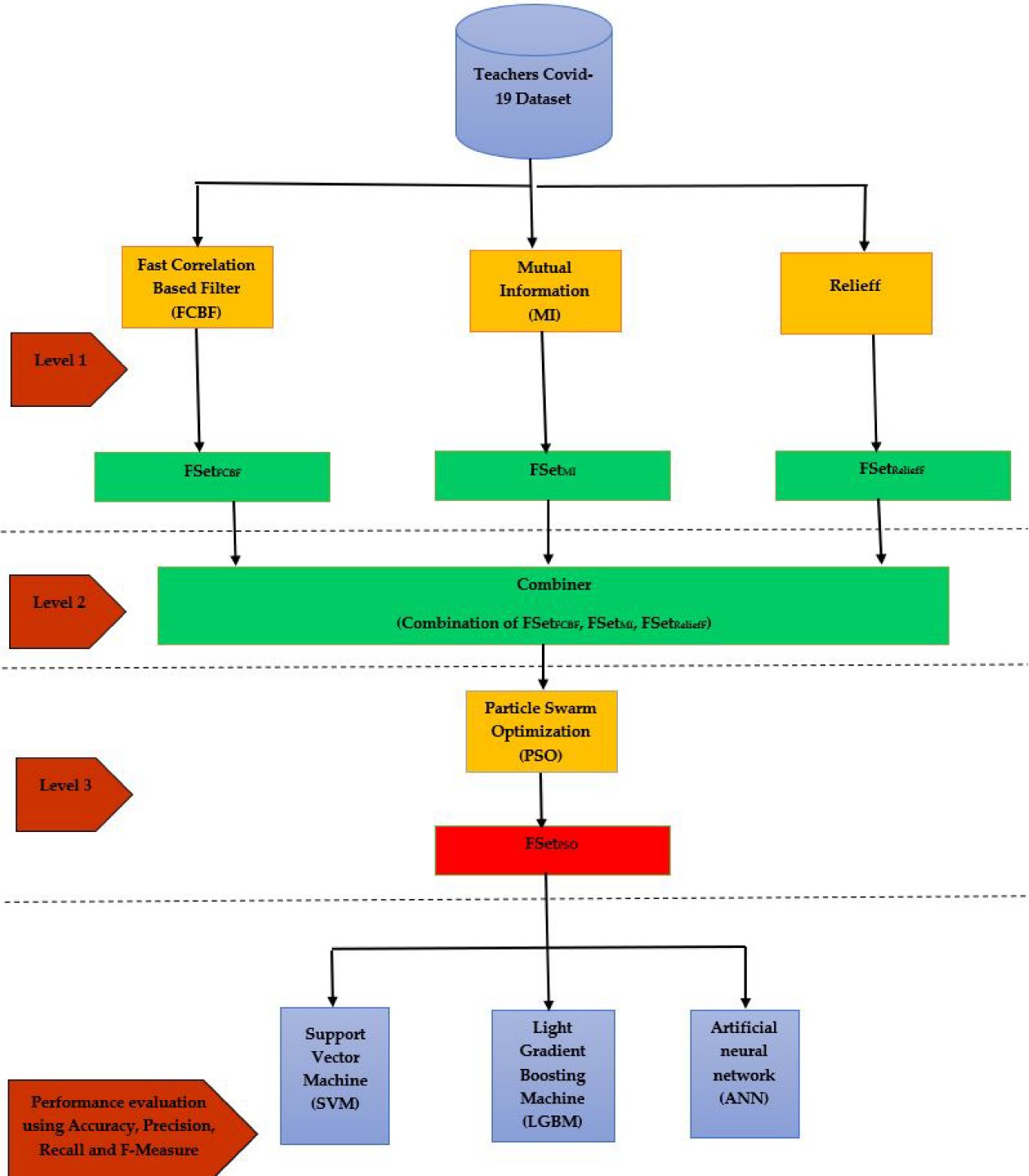

**Figure 1.** Proposed Approach for Identifying the Factors Affecting the satisfaction of Teachers during COVID-19.

### 3.1. Dataset Details

Vietnam's Benchmark dataset [26] is being used to identify the factors affecting the performance of students. The dataset is constructed by 294 Vietnamese teachers' perspectives on the teaching profession, the dataset is constructed through a questionnaire having 36 questions in it. The dataset is composed of teacher's data during online teaching and learning during COVID-19. The detail of the features of the dataset is explained in Table 2. Whereas in Table 2 TD, D, ND nor A, A, TA stands for "Totally disagree", "Disagree", "Neither disagree nor agree", "Agree", and "Totally agree".

**Table 2.** Details of Benchmark Vietnam's Teachers' Dataset.

| Feature | Feature Details | Description (Possible Values) |
|---|---|---|
| Gender | Your Gender | Male, Female, Prefer not to disclosure |
| Exp | Teaching Experience | Less than 3 years, From 3 to 5 years, From 5 to 10 years, More than 10 years |
| Degree | Teaching qualification | Diploma, BA, MA, Doctor |
| Grade_level | You are teaching students at which grade level? | Pre-K, Primary, Lower Secondary, Upper Secondary, Post Secondary |
| Subject | What subjects are you teaching? | Sciences-related, Social Sciences-related, Foreign Language, Others |
| School type | What type of school are you teaching? | Public, Private (normal), Private (bilingual/international), Continuing Education Centre, Other |
| Feel_covid | Overall, COVID-19 is affecting your health? | TD, D, ND nor A, A, TA |
| Feel_habit | COVID-19 changed your daily habit and make you tired? | TD, D, ND nor A, A, TA |
| Feel_fin | COVID-19 threatening your financial plan? | TD, D, ND nor A, A, TA |
| Income before | Your monthly income before COVID-19? (USD) | <214, 214~427, 427~641, 641~855, >855 |
| Income during | Your monthly income during COVID-19? (USD) | <214, 214~427, 427~641, 641~855, >855 |
| Income expect | What is your expected income after COVID-19? (USD) | <214, 214~427, 427~641, 641~855, >855 |
| Sup_bod | During COVID-19, you received support from the school board of management? | TD, D, ND nor A, A, TA |
| Sup_parents | During COVID-19, you received supports from the parents association? | TD, D, ND nor A, A, TA |
| Sup_union | During COVID-19, you received supports from the teacher union? | TD, D, ND nor A, A, TA |
| Sup_gov | During COVID-19, you received supports from the government? | TD, D, ND nor A, A, TA |
| Sup_none | During COVID-19, you do not receive any support? | TD, D, ND nor A, A, TA |
| ICT_before | I mastered online teaching tools before COVID-19 | TD, D, ND nor A, A, TA |
| ICT_difficult | I do not face any difficulties in online teaching during COVID-19 | TD, D, ND nor A, A, TA |
| ICT_diverse | I know many kinds of ICT platforms, tools, and applications to teach online | TD, D, ND nor A, A, TA |
| ICT_proactive | I often get to know the new technologies proactively | TD, D, ND nor A, A, TA |
| ICT_extend | I know many tools and applications more than what my school provide | TD, D, ND nor A, A, TA |
| Onl_effective | I feel that online teaching is as effective as a normal class | TD, D, ND nor A, A, TA |

**Table 2.** *Cont.*

| Feature | Feature Details | Description (Possible Values) |
|---|---|---|
| Onl_active | I feel that students are actively engaged with online sessions | TD, D, ND nor A, A, TA |
| Onl_workload | I feel that the teaching workload is much more than before COVID-19 | TD, D, ND nor A, A, TA |
| Onl_stress | I feel stressed because of online teaching | TD, D, ND nor A, A, TA |
| Ready_ICT | The ICT infrastructure of my school is ready for transformation during COVID-19 | TD, D, ND nor A, A, TA |
| Ready_teacher | The teacher capabilities of my school are ready for transformation during COVID-19 | TD, D, ND nor A, A, TA |
| Ready_policy | The policies and regulations of my school are ready for transformation during COVID-19 | TD, D, ND nor A, A, TA |
| New_ICT | During COVID-19, I have learned lots of new ICT knowledge and skills | TD, D, ND nor A, A, TA |
| New_Pedagogy | During COVID-19, I have learned lots of new pedagogical knowledge and skills | TD, D, ND nor A, A, TA |
| New_by_bod | Most of my new knowledge and skill is due to the support of my school | TD, D, ND nor A, A, TA |
| New_by_colleagues | Most of my new knowledge and skill is due to the support of my colleagues | TD, D, ND nor A, A, TA |
| New_lackoftime | I do not have proper time to elevate my profession | TD, D, ND nor A, A, TA |
| Satis_teach_learn | I am satisfied with online teaching and learning | TD, D, ND nor A, A, TA |
| Satis_life | I am satisfied with the supportiveness I received to ensure my living | TD, D, ND nor A, A, TA |
| | Pseudocode 1: Proposed Methodology | |
| Input | Vietnam Teachers_Covid-19 Dataset (VT_DS) with 35 features f1, f2, . . . ., f35 | |
| Output | Display the classification results of Accuracy, Precision, Recall, and F-Measure of different classifiers | |
| | Function Multilevel FS () | |
| 1 | Perform FS on VT_DS (having 35 features) using FCBF (FSetFCBF feature set is retrieved having best features according to FCBF feature selection) | |
| 2 | Perform FS on VT_DS (having 35 features) using MI (FSetMI feature set is retrieved having best features according to MI feature selection) | |
| 3 | Perform FS on VT_DS (having 35 features) using Relieff (FSetRelieff feature set is retrieved having best features according to Relieff feature selection) | |
| 4 | Combining FSetFCBF, FSetMI, and FSetRelieff selecting optimal features. (FSetCB is retrieved by combining three feature sets) | |
| 5 | Perform FS on FOPT using PSO (FSetFinal feature set is retrieved having best features according to PSO feature selection) | |
| 6 | Calculate the classification accuracy through various existing machine learning classifiers, e.g., SVM, LGBM, and ANN on FSetFinal. | |
| 7 | Compare the results. End | |

Pseudocode 1 explains the main working of the proposed approach, to identify the factors affecting the satisfaction of teachers on online learning during COVID-19. At the first level, the dataset is given as input to FCBF feature selection, MI feature selection, and Relieff feature selection. As a result, three feature sets are extracted named, FSetFCBF, FSetMI, and FSetRelieff with a different number of features. These three feature sets are then

combined through the combiner function. The union of three sets is performed to combine the three feature sets. The combined feature set is named FSetCB. To get the optimal features, FSetCB is given as input to PSO feature selection, and hence final FSetFinal is retrieved at the end, Whereas SVM, LSBM, and ANN machine learning algorithms evaluate the performance of the proposed feature selection method.

### 3.2. Feature Selection

Feature selection is reported to increase the efficiency of the classifier by selecting the best features. There are mainly three types of features: Filter, Wrapper, and Hybrid. Each type of feature selection has its pros and cons. Filter feature selection is fast processing and takes an account of each feature's importance and the relationship between the features through different statistical measures. As the main aim of this research is to retrieve the optimal features that affect the satisfaction of teachers in teaching and learning during COVID-19, the importance of each feature and its relationship with other features is taken into account. The four Filter feature selections are hybridized in such a way that optimal features are extracted.

FCBF (Fast Correlation Based-Filter): FCBF feature selection is based on Symmetrical Uncertainty (SU) and is proposed by Yu and Liu in 2004 [27], which is defined as the ratio between the information gain (IG) and the entropy (H). As FCBF only considers symmetric uncertainty or maximum information coefficient in measuring redundant features. However, FCBF has a problem where some relevant features are considered redundant features and removed [28]. The main reason of utilizing FCBF on COVID-19 teachers' dataset, is its checks the correlation of each of the feature and also the relevance of each of the feature with the target variable. In the Vietnam COVID-19 dataset, it is important to analyze the factors affecting the satisfaction of teachers, as FCBF considers the correlated features, it may identify the optimal factors affecting the satisfaction of teachers in COVID-19.

MI: Mutual Information (MI) is used to calculate the dependency between random features. It is an asymmetric measurement that can recognize non-linear relationships between features [29]. As it calculates the statistical dependences between the features, so MI feature selection is utilized in selecting the features affecting the satisfaction of teachers on online learning during COVID-19.

Relieff: Relieff is a feature weighting and upgraded version of the relief algorithm. It can deal with noisy, multiclass datasets with low bias. Weights are generated using Manhattan distance in Relieff instead of Euclidean distance as used by the relief algorithm [30]. Relieff can detect the conditional dependencies between the features. Below are the steps of the pseudo-code of Relief-Algorithm:

Step 1: Initialize weights by 0.
Step 2: Select an instance randomly.
Step 3: Nearest Hit; Searching for the nearest neighbor in the same class.
Step 4: Nearest Misses; Searching nearest neighbor from different classes.
Step 5: Applying weight estimation based on the values of steps 2, 3, and 4.

Relieff feature selection is used in different areas of research to select the optimal features. However, it fails to remove redundant data [31]. Furthermore, it considers the interaction between the features, which are usually ignored by FCBF [31,32]. And the main reason for utilizing Relieff in our research is its ability to rank each feature, and then selecting high-ranked features.

PSO (Particle Swarm Optimization): PSO algorithm is used to improve the algorithm optimization. It has the advantage of easy implementation and PSO parameters are easily adjustable [33,34]. It was proposed by Kennedy and Eberhart in 1995 to solve the optimization problem [35]. There are two categories of PSO in terms of feature selection, binary PSO [36] and continuous PSO [37]. For our research, we have used binary PSO due to its two-class problem compatibility.

### 3.3. Classification Models

SVM: SVM was proposed by Vapnik in 1995, it is a supervised learning technique to resolve classification and regression-related problems. The main idea behind SVM is the selection of the best hyperplane that maximizes the margin between positive and negative data points [38]. The COVID SVM classification algorithm has been used in different areas of research like in health care [39], education [40], and vice versa.

LGBT: Light gradient boosting algorithm achieves high and robust accuracy for classification and regression [41]. It has a decision tree as a base classifier. It works better than XGBoost in terms of memory and speed consumption [42].

ANN: Artificial Neural Network (ANN) connects a set of input to a set of outputs. This connection is performed by learning the training data set [43]. This is a computing method stimulated by the working of the human brain. It consists of different processing layers [44].

### 4. Results and Discussion

This section will illustrate the results of the proposed method. Figure 2 present the flow of results on the COVID-19 teacher data set, in a simpler way. Python programming language is utilized for coding the proposed approach, due to its simplicity and built-in machine learning libraries. The data set is divided into an 80:20 ratio for testing and training. Feature selection is a preprocessing technique, so the dataset is preprocessed by different feature selection techniques. Vietnam COVID-19 teachers' dataset is already labeled benchmark dataset. However, 1 to 5 is the range of satisfaction. By applying the scaling technique, we consider values below 3 as not satisfied, and above as satisfied. Min-Max scaling technique target class (feature "Satis_teach_learn") is rescaled. The main reason for using the scaling technique in our approach is to avoid the trend of machine learning weight greater values higher and smaller values as the lower values, despite the consequences of their real meanings. In the proposed work the factors affecting the teacher's satisfaction with online learning during COVID-19, are identified by multilevel feature selection. At the first level of the proposed work, the FCBF feature selection process the teacher dataset and selects the eight best factors affecting the target feature (Satis_teach_learn), then the MI feature selection algorithm is applied on COVID -19 teacher's dataset, MI feature selects 13 factors affecting the performance of teachers. Furthermore, Relieff feature selection selects 15 features from the COVID-19 teachers' dataset. At the end of the first level of the proposed work, three feature sets of different numbers 8, 13, and 15 are selected by three different feature selection algorithms.

To avoid the repetition of features, the union of three feature sets selected by FCBF, MI, and Relieff is taken at the second level of the proposed approach. A union of three feature set is performed by union combiner function, that results into a feature set with 23 features. At the third level of the proposed approach, the PSO feature selection process the COVID teacher dataset with 23 selected features. PSO identifies four main factors affecting the satisfaction of teachers during COVID-19 online learning techniques. Feature selection is performed at three levels of the proposed approach. Table 3 shows feature analysis features selected by the proposed approach. The features are selected in three levels, the top five features selected at each level of the proposed approach are presented. Furthermore, the analysis shows that New_by_bod (Most of my new knowledge and skill is due to the support of my school), and Ready_teacher (The teacher capabilities of my school are ready for transformation during COVID-19) features are most selected by different algorithms at each level of the proposed approach. This feature analysis concludes that the support of academic institutes is very crucial for teachers' satisfaction during teaching and learning in COVID-19. No doubt this dataset belongs to Vietnam teachers but during COVID-19, teachers all around the world face difficulties during online teaching.

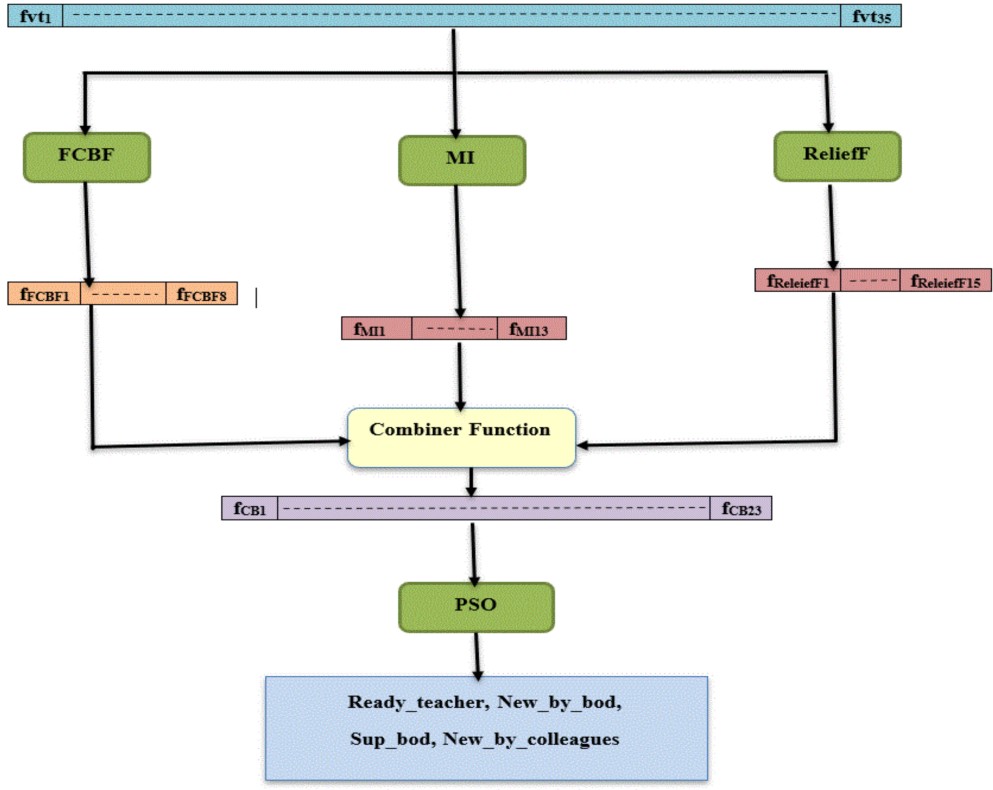

**Figure 2.** Flow of Results for Covid Teacher Dataset.

**Table 3.** Feature Selection Analysis of Proposed Approach.

| Proposed Approach | Levels | Top 5 Features |
|---|---|---|
| FCBF | | New_by_bod (Most of my new knowledge and skill is due to the support of my school), Ready_teacher (The teacher capabilities of my school are ready for transformation during COVID-19), Income expects (What is your expected income after COVID-19? (USD), Feel_fin (COVID-19 threatening your financial plan), Gender, |
| MI | Level 1 | Sup_bod (During COVID-19, you received support from the school board of management?), New_by_bod (Most of my new knowledge and skill is due to the support of my school). Sup_gov (During COVID-19, you received support from the government), New_by_colleagues (Most of my new knowledge and skill is due to the support of my colleagues), Ready_ICT (The ICT infrastructure of my school is ready for transformation during COVID-19) |
| Relieff | | Sup__none (During COVID-19, you do not receive any support?), Sup_bod (During COVID-19, you do not receive any support?), New_by_bod (Most of my new knowledge and skill is due to the support of my school), Onl_effective (I feel that online teaching is as effective as a normal class) |
| Combiner Function | Level 2 | Exp (Teaching Experience), Ready_teacher (The teacher capabilities of my school are ready for transformation during COVID-19), Income_during (Your monthly income during COVID-19?), New_by_bod (Most of my new knowledge and skill is due to the support of my school), Sup_bod (During COVID-19, you do not receive any support?) |
| Final Feature set (PSO) | Level 3 | Ready_teacher (The teacher capabilities of my school are ready for transformation during COVID-19), New_by_bod (Most of my new knowledge and skill is due to the support of my school, Sup_bod (During COVID-19, you do not receive any support?), New_by_colleagues (Most of my new knowledge and skill is due to the support of my colleagues), Income_before (Your monthly income before COVID-19?) |

Figure 2 shows a flowchart that depicts that at final stage of the proposed approach of four features, Ready_teacher (The teacher capabilities of my school are ready for transformation during COVID-19), New_by_bod (Most of my new knowledge and skill is due to the support of my school), Sup_bod (During COVID-19, you received support from the school board of management?), and New_by_colleagues (Most of my new knowledge and skill is due to the support of my colleagues) are extracted from the COVID-19 teachers dataset. The extracted features show that those teachers who get school support in learning new skills as well as get management support from their academic institution during COVID-19 are satisfied with online learning during COVID-19. Additionally, the teacher's ability to adopt the new technology for online learning plays a vital role. Furthermore, working in a cooperative environment also affects the satisfaction of teachers on learning during COVID-19.

The proposed approach is evaluated through four different evaluation measures, accuracy, precision, recall, and f-measure. Accuracy is a measure of the effectiveness of the proposed model. It shows how correctly the proposed approach identifies the instances in the right target class. Whereas recall, precision, and f-measure evaluate the proposed approach in a more detailed manner, in terms of explaining the rightly classified instances predicted by the proposed methodology.

Figure 3 presents the comparative analysis of proposed feature selection in terms of accuracy on the COVID-19 teacher dataset with other existing feature selection methods like PSO, Relieff, FCBF, and MI. Equation (1) presents the formula for accuracy.

$$\text{Accuracy} = \frac{\text{Number of teacher correcly classified}}{\text{Total number of teachers}} \tag{1}$$

The result shows that the proposed method outperforms existing feature selection methods in terms of accuracy using ANN, LGBM, and SVM machine learning algorithms. Furthermore, results show that LGBM gives more than 80% accuracy on the proposed feature selection method.

Figure 4 presents the results of a comparison between the proposed feature selection method with existing feature selection methods based on precision.

$$\text{Precision} = \frac{\text{Number of satisfied teachers identified by the proposed approach}}{\text{Total number of satisfied teachers and unsatisfied teacher classified by proposed approach}} \tag{2}$$

Analysis of results shows that ANN and LGBM show better results on the proposed feature selection method.

Figure 5 shows the results of recall on the proposed feature selection method and their comparison with existing feature selection methods. The formula for calculating the recall is explained as follows

$$\text{Recall} = \frac{\text{Number of satisfied teachers classfied by the proposed approach}}{\text{Total number of satisfied teachers}} \tag{3}$$

Similarly, for accuracy and precision, the recall results of the proposed work for identifying the factors affecting the satisfaction of teachers during COVID-19 online learning outperform existing feature selection techniques.

Figure 6 shows that the proposed feature selection method performs the best result on F-measure performance than other existing feature selection methods. Whereas the formula for calculating f-measure is presented through Equation (4):

$$\text{F} - \text{measure} = \frac{2 \times (\text{Precision} \times \text{Recall})}{\text{Precsion} + \text{Recall}} \tag{4}$$

More than 80% of the recall value of the proposed feature selection is noticed using the LGBM machine learning algorithm.

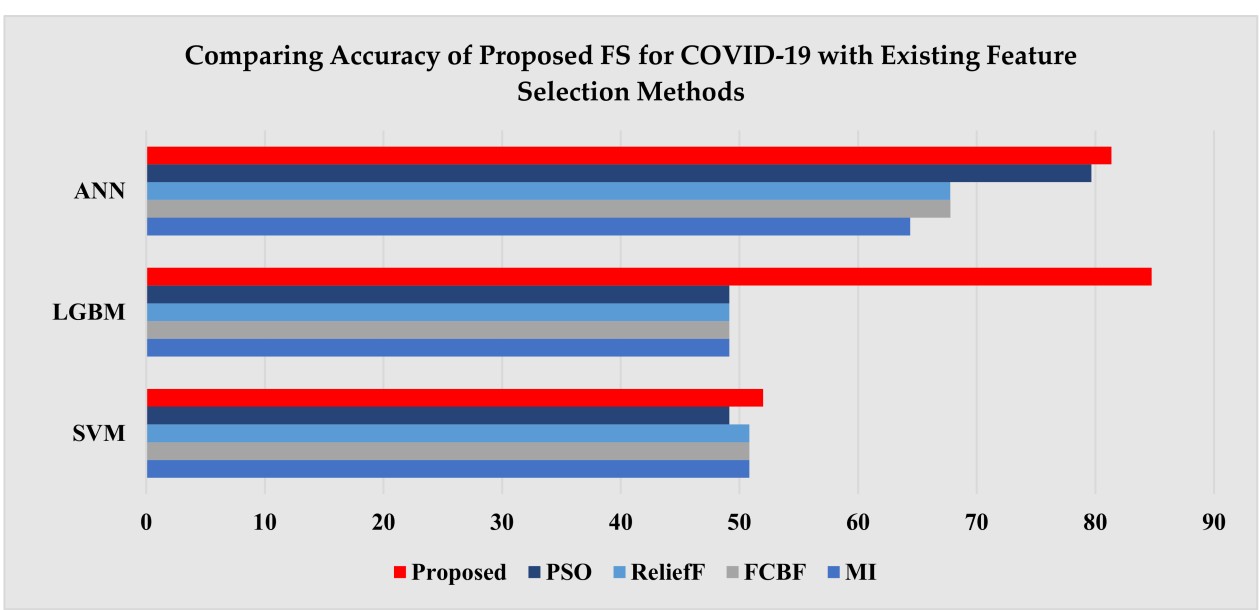

**Figure 3.** Accuracy Comparison of Proposed FS for Teachers Satisfaction in COVID-19 with Existing Feature Selection Methods.

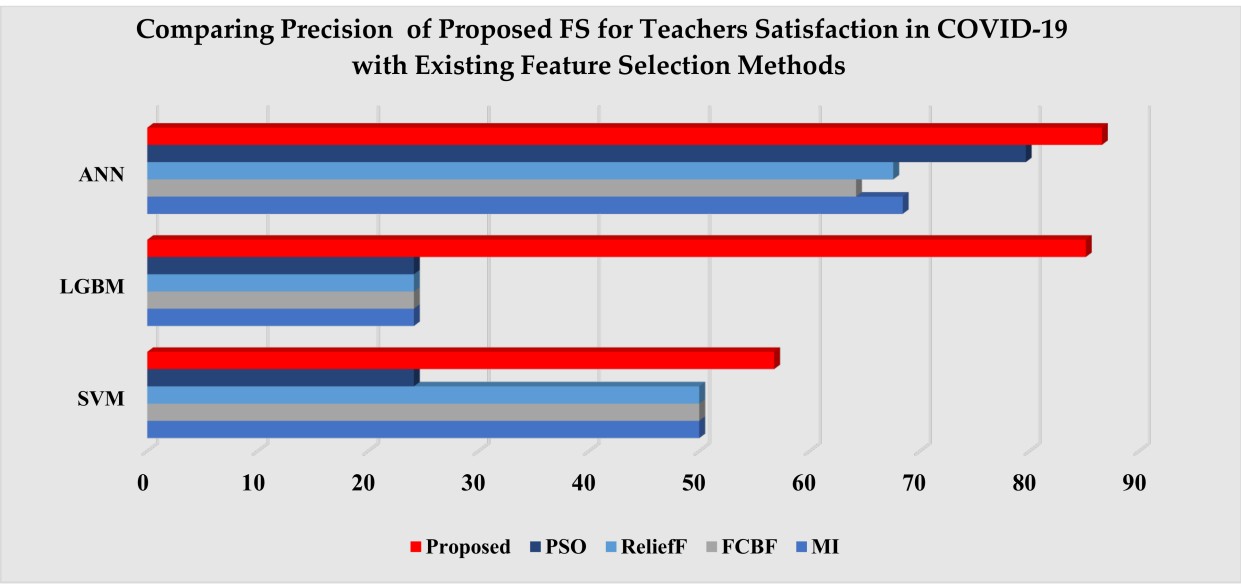

**Figure 4.** Comparing Precision of Proposed FS for Teachers Satisfaction in COVID-19 with Existing Feature Selection Methods.

The result shows that the proposed method outperforms other existing feature selection methods in terms of accuracy, precision, recall, and f-measure. Different machine learning algorithms are applied on the selected feature dataset of the proposed approach however performance of LGBM is observed outclass, among others. Furthermore, the proposed feature selection results that the satisfaction of teachers during COVID-19 is affected by the support of their academic institution in learning new technologies and the help of academic institution management for online teaching during COVID-19. As different factors appear related to the satisfaction of teachers, one of them is the support from other colleagues in COVID-19. The proposed approach figures out the main factors affecting the satisfaction of teachers during COVID-19. As the teachers are the main stakeholder of education, academic administration should be helpful for them in teaching and providing them all facilities for learning the new skills. This will help in improving the quality of

education during the COVID-19 pandemic situation. Moreover, teachers must have the ability to grasp the new technology for online teaching and learning during COVID-19 with full interest.

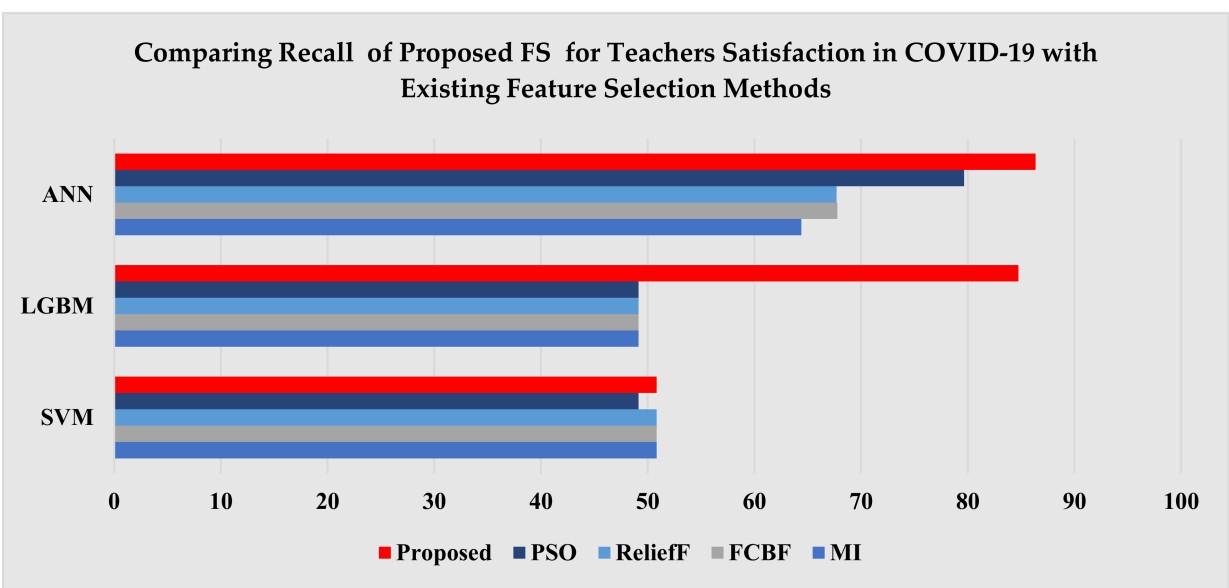

**Figure 5.** Recall Comparison of Proposed FS for Teachers Satisfaction in COVID-19 with Existing Feature Selection Methods.

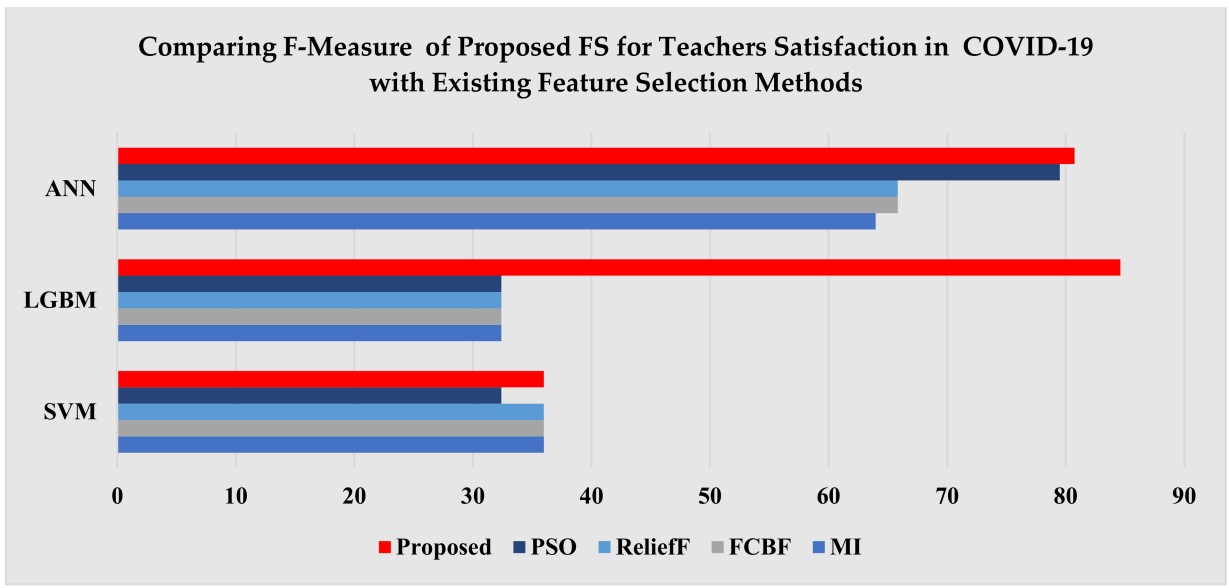

**Figure 6.** F-Measure Comparison of Proposed FS for Teachers Satisfaction in COVID-19 with Existing Feature Selection Methods.

## 5. Conclusions

In this paper, we proposed a multilevel feature selection for identifying the factors affecting the satisfaction of teachers on online teaching and learning during COVID-19. The results show that support of the academic institution to teachers in COVID-19 can play a vital role in improving the quality of education in this pandemic. If teachers get support from institutions in learning new technologies for conducting online learning, the teachers have a better way of teaching and learning. Just as the working environment matters a lot, the same can be said about online teaching and learning during COVID-19. Colleagues can guide each other in making online teaching and learning better during COVID-19, by sharing knowledge, experiences, and techniques. One of the important

features identified by the proposed approach is the readiness of teachers to transform according to new techniques of teaching during COVID-19. Furthermore, proposed feature selection performance is evaluated through four evaluation measures, accuracy, precision, and recall. The proposed approach outperforms existing feature selection methods in all four evaluation measures. In the future, we will hybridize different feature selection algorithms for extracting optimal features with better accuracy. Furthermore, as the proposed approach utilized the Vietnam COVID-19 teachers' dataset for evaluation, in the future more teachers' COVID-19 datasets can be utilized to analyze the factors affecting the teacher's satisfaction during online teaching and learning in COVID-19.

**Author Contributions:** A.S., R.H. and M.Z. performed the experiment, analyzed the data and wrote the paper K.S.Q. and O.A. performed comparative analysis. M.I., A.G., R.T. and M.A.H. were responsible for project management. A.A., S.K.A. and F.A. were responsible for editing, resource management and data visualization. A.K.A. and Z.A. edited and rewrite the paper. All authors have read and agreed to the published version of the manuscript.

**Funding:** This research was funded by the AGH University of Science and Technology, grant No. 16.16.120.773 for APC payment.

**Acknowledgments:** The authors acknowledge the support from the Ministry of Education and the Deanship of Scientific Research, Najran University, Kingdom of Saudi Arabia, under code number NU/MID/18/030.

**Conflicts of Interest:** The authors declare that they have no conflict of interest to report regarding the present study.

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
