# Peer review of "Analyzing the Features Affecting the Performance of Teachers during Covid-19: A Multilevel Feature Selection"

_electronics, doi:10.3390/electronics10141673_

Round 1

Reviewer 1 Report

While I appreciate the content of the study, there are a number of concerns in the way it is presented: 

1) Some of these are language related where specific words are used, which do not seem appropriate. This may be due to the use of direct translation, but means that words are not suitable to the context they are used. These have largely been marked on the copy either through a circle around, a question mark or squiggles underneath (which often also means this needs to be clarified). One such strange word uses is: "understudies", which needs to at least be defined at the start of the text, as to what I means. There are issues with abbreviations use and so on. In some areas, the poor language hinders proper understanding, and so this needs to be addressed with urgency to make this text understandable. The text also includes several bits of repetition - that needs to go a well written paper doesn't need repetition ("say it once, say it well" should be the motto). 

2) There seem to be 2 aspects to the study: a) the data being gathered from a questionnaire and then b) the feature extraction methods used. In many places the feature extraction methods take priority, but then suddenly there are conclusions about the dataset itself, which are not substantiated or even shown in the results, but just concluded. The latter is inappropriate. If there are two aspects, then they need to both be more clearly described which would then allow conclusions about the data gathered as well, while this is currently quite inappropriate.

3) The result section needs serious work as it doesn't explain the tests and outcomes clearly. The graphs are partly confusing and seem repetitive, and it is also not clear what they add in/as in display based on what you wanted to achieve. Coming back to point 2, it also seems that the feature extraction needs to be more clearly separated or clarified as to which data it relates to (if there is a relation, as it is most likely that all data was used). Earlier on, when methods used are described (Methodology section) there needs to be more justification as to why these specific methods were chosen. There are statements as to which items are chosen and their "characteristics", but why were they chosen for this setting - as in why are they even closely suitable. 

4) References should be consistently formatted. Author names are formatted differently with regards to abbreviating them (first reference versus later ones), while this should be consistent and the use of "et al" should be restricted and only be used after a number of author names have already been provided, only showing one followed by et all is not suitable for a reference list at the end. 

Please note that there is a copy with extensive markup provided, which are all items to be addressed, they go into more detail about some of the above mentioned points. I would hope that the write up can be deciphered, as it is key to provide the necessary clarity into the paper to make it understandable. I would also recommend that you get this read by a native speaker before you submit it again. 

Author Response

Dear Reviewer thank you for your valuable comment we tried our best to address all the comments mentioned.

Point 1: Some of these are language related where specific words are used, which do not seem appropriate. This may be due to the use of direct translation, but means that words are not suitable to the context they are used. These have largely been marked on the copy either through a circle around, a question mark or squiggles underneath (which often also means this needs to be clarified). One such strange word uses is: "understudies", which needs to at least be defined at the start of the text, as to what I means. There are issues with abbreviations use and so on. In some areas, the poor language hinders proper understanding, and so this needs to be addressed with urgency to make this text understandable. The text also includes several bits of repetition - that needs to go a well written paper doesn't need repetition ("say it once, say it well" should be the motto). 

Response 1: Introduction and related work is rewrite to address the comments. Needful is done. Regards

Point 2: Some of these are language related where specific words are used, which do not seem appropriate. This may be due to the use of direct translation, but means that words are not suitable to the context they are used. These have largely been marked on the copy either through a circle around, a question mark or squiggles underneath (which often also means this needs to be clarified). One such strange word uses is: "understudies", which needs to at least be defined at the start of the text, as to what I means. There are issues with abbreviations use and so on. In some areas, the poor language hinders proper understanding, and so this needs to be addressed with urgency to make this text understandable. The text also includes several bits of repetition - that needs to go a well written paper doesn't need repetition ("say it once, say it well" should be the motto). 

Response 2: Comments are addressed.

Point 3: The result section needs serious work as it doesn't explain the tests and outcomes clearly. The graphs are partly confusing and seem repetitive, and it is also not clear what they add in/as in display based on what you wanted to achieve. Coming back to point 2, it also seems that the feature extraction needs to be more clearly separated or clarified as to which data it relates to (if there is a relation, as it is most likely that all data was used). Earlier on, when methods used are described (Methodology section) there needs to be more justification as to why these specific methods were chosen. There are statements as to which items are chosen and their "characteristics", but why were they chosen for this setting - as in why are they even closely suitable. 

Response 3: Explanation added in methodology.

Point 4: References should be consistently formatted. Author names are formatted differently with regards to abbreviating them (first reference versus later ones), while this should be consistent and the use of "et al" should be restricted and only be used after a number of author names have already been provided, only showing one followed by et all is not suitable for a reference list at the end. 

Response 4: References are edited.

Dear reviewer comments mentioned in attached file are addressed, tried best to meet the requirements mentioned. Thank you again.

Kind Regards

Reviewer 2 Report

The paper is related to the evaluation of COVID-19 related features among teachers.
The motivation, aim, and contributions of the paper are presented, although in the whole introduction section there is no information about the character of analyzed features. What is more, in the introduction section there is no information about the type of teachers to which this study is addressed – are they university teachers or primary school teachers? There is an essential difference and the paper motivation should cover this issue.
The literature review is rather weak as it does not cover similar research. What is more, none of the cited works is described in a more exhausting way to discuss obtained results and their influence on the author’s work.

The dataset applied in the study was taken from the paper of Du et. al and it is a dataset of Vietnamese teachers. This is a very important issue not mentioned before section 3.1 and the overall results should be discussed in the context of teaching standards in Vietnam before and during COVID-19. What is more, the results should be interpreted also in regards to these standards.
There should be presented also other research conducted using this dataset. 
Some more information about the data should be also discussed. Are the subsets of different types of teachers equally distributed (especially degree, grade level, school type, subject)? 

The procedure presented in the Pseudocode 1 should be explained in more detail. Have you used any normalization? How the features were pre-processed? Have been any outliers detection procedure?

In my opinion applied methods should be described in another section that the feature selection results are presented. What is more, section 3.2 does not contain sufficient information about the parameters of applied methods, the whole processing pipeline is described chaotically and inaccurately.
Section results and discussion should be separated.
The authors present results of classification, but they don’t give details of this classification procedure (how many classes, how these classes were defined, how the train-test split was performed). 
There is no discussion section that should cover the interpretation of the presented results and the most important features. 

What is more, some statistical analysis would be also valuable here.

Authors should put more attention to spelling:
Some spelling and stylistic errors: every filed of life (l. 28); LGBM – misspelling full name (l. 40); LGBT -> LGBM?? (l. 84); build up their abilities, abilities (l.100, repetition); Line 125: Students,(…), are questions

Line 102 why in this sentence there is a phrase written in capital letters?
Line 114: who is Daniel? I cannot see any citation in this sentence

Some problems related to references, for example in 8, where there is no separation between author and title

Author Response

Dear Reviewer thank you for your valuable comments, we tried our best to address all the comments mentioned.

Point 1: The paper is related to the evaluation of COVID-19 related features among teachers.
The motivation, aim, and contributions of the paper are presented, although in the whole introduction section there is no information about the character of analyzed features. What is more, in the introduction section there is no information about the type of teachers to which this study is addressed – are they university teachers or primary school teachers? There is an essential difference and the paper motivation should cover this issue.

Response 1:  Introduction is rewritten, Paper focuses on vitenam benchmark dataset of teachers of different school and college levels.

Point 2: The literature review is rather weak as it does not cover similar research. What is more, none of the cited works is described in a more exhausting way to discuss obtained results and their influence on the author’s work.

Response 2: Added relevant literature.

Point 3: The procedure presented in the Pseudocode 1 should be explained in more detail. Have you used any normalization? How the features were pre-processed? Have been any outliers detection procedure?

Response 3: Please provide your response for Point 2. (in red)

Point 4: In my opinion applied methods should be described in another section that the feature selection results are presented. What is more, section 3.2 does not contain sufficient information about the parameters of applied methods, the whole processing pipeline is described chaotically and inaccurately.

Response 4: Dear sir/madam explanation is added in the result section.

Point 5: Section results and discussion should be separated.

Response 5 ;Results are explained

Point 6: The authors present results of classification, but they don’t give details of this classification procedure (how many classes, how these classes were defined, how the train-test split was performed).

Response 6: Mainly two classes in dataset. Teachers satisfaction and not satisfaction during covid-19 on online learning. Data is splitted in a ratio of 80:20 of testing and training.

Point 7: There is no discussion section that should cover the interpretation of the presented results and the most important features. 

Response 7: Tried best to rewrite  some explanation.

Point 8: Authors should put more attention to spelling:
Some spelling and stylistic errors: every filed of life (l. 28); LGBM – misspelling full name (l. 40); LGBT -> LGBM?? (l. 84); build up their abilities, abilities (l.100, repetition); Line 125: Students,(…), are questions

Response 8: needful done

Point 9: Line 102 why in this sentence there is a phrase written in capital letters?
Line 114: who is Daniel? I cannot see any citation in this sentence.

Response 9: rewritten, needful done

Point 10: Some problems related to references, for example in 8, where there is no separation between author and title.

Response 10: references are added.

Dear reviewer thank you again to give such valuable comments for the improvement of our paper. we tried our best to address to your valuable comments.

Kind Regards

Round 2

Reviewer 1 Report

While it is good to see most of the content related items addressed during your review. There are still some outstanding issues that haven't been addressed with regards to references and language. On the attached file you will find corrections for language (words that need to be removed, "s's" that need to be added/removed, among other items which require clarification e.g. use of "it" or words that don't make any sense at all). I recommend that those get addressed and as probably already mentioned, you get the paper read by a native speaker. 
The comment on the reference naming convention at the end didn't seem to get addressed. Naming should always start with surname or always with abbreviation, so e.g. currently you have Rees, P. and N, Seaton, which is inconsistent, and should be: Rees, P. and Seaton, N., This applies throughout, and as mentioned with version 1, you should only have an "et all" after mentioning at least 3-4 author names, not after one. 

Author Response

Dear Reviewer

Thank you for giving me valuable comments for  the improvement of my article. I tried my best to address all issues.

Reviewer 2 Report

The introduction is still short and it does not give enough insights into the discussed issue. There is also no description of changes related to COVID-19 in Vietnam’s education. Do authors assume that results might be generalized to other countries? It should be explained.

Some of my previous remarks/questions are not addressed (for example the question about pre-processing procedure).

If the feature analysis is the aim of the paper, where is it? It would be necessary to rank features and discuss them as well as to analyze the feature impact on the classification results.
What is more, the presentation of features revealed by different feature selection methods should be added.

As I understand there is a two-class classification performed. What were these classes, satisfied and not satisfied? How the class labels were set? Which feature was applied in order to determine the belonging of observations to the classes? How many observations were in the particular classes?

There are still spelling errors, for example:
“Different factors are ident in recent studies that ma effect” or  “is its checks”

Author Response

Dear Reviewer

Thank you for giving me such valuable comments for the improvement of my article. I tried my best to follow the points indicated by respected reviewer.

Point 1: The introduction is still short, and it does not give enough insights into the discussed issue.

Response 1: Needful Done

Point 2: There is also no description of changes related to COVID-19 in Vietnam’s education. Do authors assume that results might be generalized to other countries? It should be explained.

Response 2 Needful done  

Point 3: Some of my previous remarks/questions are not addressed (for example the question about pre-processing procedure).

Response 3: The Vietnam teachers COVID-19 dataset contains no missing value. However, scaling technique is utilized to rescale the target class for better classification results.

Point 4: If the feature analysis is the aim of the paper, where is it? It would be necessary to rank features and discuss them as well as to analyse the feature impact on the classification results.
What is more, the presentation of features revealed by different feature selection methods should be added.

Response 4: Needful done along with an explanation.

Proposed Approach

Levels

Top 5 Features

FCBF  

Level 1

New_by_bod (Most of my new knowledge and skill is due to the support of my school),

Ready_teacher (The teacher capabilities of my school is ready for transformation during COVID-19), Income expects (What is your expected income after COVID-19? (USD), Feel_fin (COVID-19 threatening your financial plan), Gender,

MI

Sup_bod (During COVID-19, you received supports from school board of management?), New_by_bod (Most of my new knowledge and skill is due to the support of my school.

) , Sup_gov (During COVID-19, you received supports from the government), New_by_colleagues(Most of my new knowledge and skill is due to the support of my colleagues), Ready_ICT (The ICT infrastructure of my school is ready for transformation during COVID-19),

Relieff

Sup__none(During COVID-19, you do not receive any support?), Sup_bod (During COVID-19, you do not receive any support?), New_by_bod (Most of my new knowledge and skill is due to the support of my school), Onl_effective(I feel that online teaching is as effective as normal class)

Combiner Function

Level 2

Exp (Teaching Experience)

, Ready_teacher (The teacher capabilities of my school is ready for transformation during COVID-19), Income_during (Your monthly income during COVID-19?), New_by_bod

(Most of my new knowledge and skill is due to the support of my school),

 Sup_bod (During COVID-19, you do not receive any support?)

Final Feature set(PSO)

Level 3

Ready_teacher (The teacher capabilities of my school is ready for transformation during COVID-19),New_by_bod (Most of my new knowledge and skill is due to the support of my school, Sup_bod (During COVID-19, you do not receive any support?),New_by_colleagues (Most of my new knowledge and skill is due to the support of my colleagues), Income_before (Your monthly income before COVID-19?)

Point 5: As I understand there is a two-class classification performed. What were these classes, satisfied and not satisfied? How were the class labels set? Which feature was applied in order to determine the belonging of observations to the classes? How many observations were in the particular classes? 

 Response 5: Vietnam COVID-19 teachers’ dataset is already labeled benchmark dataset. However, 1 to 5 is the range of satisfaction. By applying the scaling technique, we consider values below 3 as not satisfied, and above as satisfied. Min-Max scaling technique target class (feature “Satis_teach_learn”) is rescaled. The main reason for using the scaling technique in our approach is to avoid the trend of machine learning weight greater values higher and smaller values as the lower values, despite the consequences of their real meanings. The proposed approach gives around 85% accuracy, which shows that around 247 teachers are classified as satisfied, whereas 47 teachers are classified as unsatisfied.

Point 6: There are still spelling errors, for example:
“Different factors are ident in recent studies that ma effect” or “is its checks”

Response 6: Needful Done

Round 3

Reviewer 2 Report

My remarks have been addressed